# Cellular Mechanosensitivity: Validation of an Adaptable 3D-Printed Device for Microindentation

**DOI:** 10.3390/nano12152691

**Published:** 2022-08-05

**Authors:** Giulio Capponi, Martina Zambito, Igor Neri, Francesco Cottone, Maurizio Mattarelli, Massimo Vassalli, Silvia Caponi, Tullio Florio

**Affiliations:** 1Dipartimento di Fisica e Geologia, Università di Perugia, 06100 Perugia, Italy; 2Sezione di Farmacologia, Dipartimento di Medicina Interna, Università di Genova, 16132 Genova, Italy; 3James Watt School of Engineering, University of Glasgow, Glasgow G12 8LT, UK; 4Istituto Officina dei Materiali, Italian National Research Council (IOM-CNR), Unit of Perugia, c/o Department of Physics and Geology, University of Perugia, Via A. Pascoli, 06123 Perugia, Italy; 5Istituto di Ricovero e Cura a Carattere Scientifico, Ospedale Policlinico San Martino, 16132 Genova, Italy

**Keywords:** mechanosensitivity, mechanobiology, mechanotransduction, 3D printing, piezo1

## Abstract

Mechanotransduction refers to the cellular ability to sense mechanical stimuli from the surrounding environment and convert them into biochemical signals that regulate cellular physiology and homeostasis. Mechanosensitive ion channels (MSCs), especially ones of Piezo family (Piezo1 and Piezo2), play a crucial role in mechanotransduction. These transmembrane proteins directly react to mechanical cues by triggering the onset of an ionic current. The relevance of this mechanism in driving physiology and pathology is emerging, and there is a growing need for the identification of an affordable and reliable assay to measure it. Setting up a mechanosensitivity assay requires exerting a mechanical stimulus on single cells while observing the downstream effects of channels opening. We propose an open-hardware approach to stimulate single adherent cells through controlled microindentation, using a 3D-printed actuation platform. We validated the device by measuring the mechanosensitivity of a neural mice cell line where the expression level and activity of Piezo1 were genetically and pharmacologically manipulated. Moreover, this extremely versatile device could be integrated with different read-out technologies, offering a new tool to improve the understanding of mechanotransduction in living cells.

## 1. Introduction

Mechanosensation is the process by which cells sense their physical environment, responding to forces [1] and probing the mechanical properties of the tissue they are embedded in [2], as well as adapting to hydrostatic pressure [3], shear flow [4], or stretch fields induced by contraction and peristalsis [5]. Mechanosensitive ion channels play a central role in this process, directly translating physical stimuli into biochemical signals and triggering the downstream transduction cascade [6,7]. During 2010, Patapoutian and collaborators discovered a new family of inherently mechanosensitive ion channels that demonstrate exquisite sensitivity: the Piezo family, including human Piezo1 and Piezo2 [8]. After their discovery, these channels were extensively studied and demonstrated central involvement in a multitude of physiological and pathological processes such as bone remodelling [9,10], gastrointestinal tract function [5], and obesogenic adipogenesis [11]. The 2021 Nobel Prize in physiology and medicine was jointly awarded to Ardem Patapoutian for this disruptive discovery, and since then researchers have started dissecting the mechanism of action of these channels. Nevertheless, many details of piezo-driven mechanosensation are still largely unexplored and poorly understood.

There is currently a clear mismatch between the physiological relevance of mechanosensation and the availability of suitable cellular assays to assess it as a function of the specific conditions of interest. The study of mechanosensation at the single cell level requires the ability to integrate the ability to apply a (controlled) mechanical stimulus and measure the induced physiological response triggered by the opening of the channels in the same set-up. The gold standard to study mechanosensation is based on using patch-clamp electrophysiology to measure the ionic current and a mix of mechanical methods to stimulate single cells, such as poking with a glass needle [12] or applying an external flow or negative pressure to the patched membrane [13]. This approach is a powerful tool to dissect the biophysics of the channels, with a resolution that can achieve single-channel sensitivity [14]. Nevertheless, patch-clamp electrophysiology is a complex and delicate technique that requires an experienced user and cannot be adopted as an every-day assay in most research laboratories [15].

More recently, calcium imaging has been suggested as a read-out method for monitoring piezo-mediated cellular mechanosensing. Piezo1 and Piezo2 are non-selective cation channels. Among others, their gating is associated with the onset of an inward flow of calcium ions [Ca^2+^] that increases the cytosolic concentration and can be monitored using fluorescence reporters. The calcium concentration rise associated with the opening of a few piezo channels cannot be appreciated with standard fluorescence-based approaches, and the intracellular localization of opening events might only be addressable using specialized chimerics as recently proposed, utilising constructs where the fluorescent reporter is directly linked to the channel [16]. Nevertheless, calcium imaging can be very effective when monitoring whole-cell responses. The primary calcium intake (associated with the opening of the channel) is normally very small and fast, but most cells actively respond to this event by activating a biochemical cascade that leads to the opening of intracellular stores and a secondary calcium intake to the cytosol that results in a much larger fluorescence increase spanning seconds [17,18]. This phenomenon has been used to monitor the response of single cells to mechanical stimulation, either applying a controlled force on the cell with an atomic force microscope [19] or via remote stimulation through acoustic waves [20,21]. Calcium imaging is available in many research laboratories and only requires a research-grade fluorescence microscope to be implemented. Nevertheless, the proposed mechanical stimulation methods are either very complex to set up or expensive and rarely found in a standard biology lab.

Here we propose an open-hardware approach to develop a single mechanosensitivity assay that can be easily implemented using low-cost 3D-printed components in combination with a calcium imaging set-up. We characterised the actuator, suggesting a calibration procedure to achieve the required spatial resolution in the displacement of the probe without the need for expensive motors. The system was tested on a cell line with controlled expression levels of Piezo1, demonstrating the suitability of the approach for studying mechanosensation and possibly addressing the pharmacology of mechanosensitive ion channels.

## 2. Materials and Methods

### 2.1. Cell Culture

A mouse immortalized Mes-c-myc A1 (hereafter A1) cell line received from Prof. Colucci (University of Campania, Caserta, Italy) was obtained as previously described from mouse embryonic mesencephalon primary culture [22]. The A1 cell phenotype was previously characterised, wherein it was found that upon differentiation, the cells retain neuronal features [23]. The cells were cultured in DMEM/F12 (Gibco-BRL, Milan, Italy) supplemented with 10% FBS (Invitrogen, Waltham, MA, USA). In order to over-express Piezo1, the A1 cell line was transfected with a human-Piezo1-GFP plasmid (donated by Prof. Boris Martinac at the Victor Chang Cardiac Institute, Sydney, Australia) using Lipofectamine LTX (Invitrogen). As this plasmid has the same antibiotic resistance as our cell line, we performed a double transfection with an empty plasmid that carried puromycin resistance. Then a first antibiotic selection and a second selection based on the GFP fluorescence of the Piezo1 plasmid were performed. Green clones were analysed using real-time PCR under specific human Piezo1 primer selection to confirm the over-expression of Piezo1 (Appendix A).

### 2.2. Calcium Imaging—Fluorescence Microscopy

For Ca^2+^ concentration imaging via fluorescence microscopy, the ratiometric calcium indicator fura2 was used [24]. In particular, Fura-2-acetoxymethyl ester (Fura-2 AM) is a membrane-permeable derivative of fura-2, which allows for intracellular imaging of Ca^2+^ concentration. Since the addition of the AM group makes the fura2 molecule lipophilic, after membrane-permeable Fura-2 AM crosses the cell membrane and is inside the cell, cellular esterase removes its acetoxymethyl group, trapping the Ca^2+^ sensitive fura2 inside the cell [25]. The A1 cells were plated in 22 mm glasses (Neuvitro, Camas, WA, USA), at a proper density to be 50% confluent, in order to avoid artefacts due to excessive cell-to-cell interactions. After 24 h, the medium was changed and loaded with Fura-2 AM (1 μM) (Abcam, Cambridge, UK). After 30 min at a temperature of 37 °C, the medium was changed with Locke buffer (HEPES10 mM pH 7.4, NaCl 150 mM, KCl 5.5 mM, CaCl_2_ 1.5 mM, MgSO_4_ 1.2 mM, glucose 10 mM) for 20 min at room temperature to avoid intracellular compartmentalisation [26] and then washed with the same balanced salt solution buffer. The fluorescence signal was collected using a dual excitation scheme, collecting the green fluorescence intensities at 510 nm after illuminating either at 340 nm or 380 nm, indicated as F_340_ and F_380_, respectively. The various components of the system (LEDs, camera, acquisition unit) were triggered using a bespoke image recording system (FAB Crea, Genoa, Italy). The fluorescence ratio FR = F_340_/F_380_ was highly independent from the concentration of the dye and was used as a reliable quantitative measure of the intracellular Ca^2+^ concentration over a broad range (1 nM–10 μM) [24]. The FR value was normalized to the initial resting baseline FR_0_, hereafter indicated as R = FR/FR_0_. A transient peak in the R signal is typical of Ca^2+^ entry induced by activation of mechanosensitive channels. The external trigger (i.e., the mechanical stimulation) induces a Ca^2+^ influx and thus the rapid increase of the fluorescence ratio signal R. By rapidly inactivating mechanosensitive ion channels [27], the cell re-equilibrates calcium homoeostasis through ATP-driven pumps [28]. On the contrary, when the cellular membrane is damaged, the pumps can no longer contrast the chemical gradient, the intracellular concentration equilibrates with that of the medium and the cell dies, resulting in an R plateau. The exposure time for both the 340 nm and 380 nm excitation wavelengths was 220 ms, resulting in a time resolution of 440 ms for ratiometric imaging. Specimens were visualised through a 40× objective (Nikon Fluor, oil immersion, NA = 1.3) mounted on an inverted microscope (Nikon Diaphot 300). At least 30 cells per field of view were analysed across three independent experiments. The Ca^2+^ images were processed with ImageJ Fiji software [29] (ROI tool). The WT A1 (*n* = 33) and A1-Piezo1 (*n* = 28) were tested using a Locke buffer. Furthermore, the control groups of A1 WT (*n* = 16) and A1-Piezo1 (*n* = 17) cells were observed in Ca^2+^ free Locke solution (Locke w/o CaCl_2_ + 1 mM EDTA) to evaluate the contribution of intracellular calcium stores. A group (*n* = 15) of A1-Piezo1 cells were pre-incubated with mechanotoxin GsMTx4 (30 min, 20 μM) and tested to assess whether the observed signals were originated by mechanosensitive opening of the channels.

### 2.3. Digital Holography

To select the ideal protocol for cell stimulation, an estimate of the cell thickness is required. We characterised the morphology of the A1 cells via digital holography microscopy (HoloMonitor^TM^ M3, Phase Holographic Imaging AB, Lund, Sweden). This technique provided a map of the optical thickness of the cells, which is proportional to the real thickness for barely homogeneous cells [30]. The analysis was performed using HoloStudio^TM^ 2.0 software.

### 2.4. Glass Probe

A glass cell-indenting probe was fabricated as previously described [31,32] by pulling a borosilicate glass tube with a 1.2 mm outer diameter (Heka PIP-6 pipette puller: T_A_ = 940 °C, T_B_ = 1200 °C, spacer disk #7), obtaining a final tip diameter of 1–2 μm. The micropipette was further fire-polished using a microforge (List Medical L/M-CPZ 101) for ~15 s to smooth the tip, obtaining a final diameter of 2–3 μm.

### 2.5. Linear Translation Stage

The bespoke glass probe (see Section 2.4) was fixed on a pipette holder, positioned with an angle ranging from 45° to 65° with respect to the horizontal plane and connected to the 3D-printed micromanipulation stage (OpenFlexure Block Stage [33]). This stage ensured a 1.5 × 1.5 × 1.5 mm^3^ range of motion and resolution on a micrometric scale (Figure 1a). The drawings for the system were provided under an open licence [34] and were printed using a fused deposition modelling (FDM) 3D printer (Ultimaker3, Ultimaker B.V., Utrecht, Netherlands) and polylactic acid (PLA). The actuator was further adapted for integration into the fluorescence microscopy set-up (Figure 1b). The translation block stage was automated for Z-axis displacements using a 28BYJ-48 stepper motor (ELEGOO, Shenzhen, China) modified to operate in a bipolar configuration [35]. The X and Y direction were used for the initial positioning, and thus manually actuated by the respective knobs. To provide a simple motor controller along the Z-axis, the stepper motor controlling the probe displacements was driven by an Arduino Uno board combined with an A4988 driver. The open-source firmware Universal G-Code Sender platform [36] enabled bidirectional indentation movements through the sending of simple g-code instructions.

### 2.6. Interferometry

The motion calibration of the stage was performed by driving the motors in a trajectory similar to the one we used in the experiments (a saw-tooth pattern) and measuring the real displacement along the axis of interest with an interferometric sensor (IDS 30310, Attocube, Haar, Germany). To perform this measurement, three retroreflectors were positioned on the moving platform of the stage and aligned with the three laser beams of the interferometric unit (IR laser sources, wavelength λ = 1530 nm). Measuring the three distances in parallel allowed for calculation of the axial motion, together with pitch and yaw angles. Aiming to calibrate the motion step along the X, Y, and Z axes, the lasers–reflectors geometry was arranged in three orthogonal orientations, one for each direction.

### 2.7. Microindentation Assay

As a preliminary result, we report the protocol we developed to measure cellular mechanosensitivity. The layout adopted for the mechanosensation experiments is sketched in Figure 2.

-Positioning the focus plane of the microscope on the upper surface of the coverslip, i.e., the adhesion plane of the cells, we indicated the position of the Z motor in which the glass-tip is in focus as Z = 0 μm.-The glass-tip was moved away from this position by a distance of 20 μm. This point was taken as the starting position for all successive indentations (h_0_ = 20 μm). -The investigated cells were positioned to have their central part (the nucleus) under the tip of the microindenter. This configuration is represented in the left panel of Figure 2. -To measure the indentation depth corresponding to cell activation, a train of 6 consecutive and progressive indentations was acquired for each cell. During the indentation train, the tip was moved towards the cell by a controlled distance D and further retracted to the rest position (h_0_ = 20 μm). The first travelled distance was set at D = 8 μm (corresponding to 12 μm above the coverslip surface) and in every successive movement the travelled distance was increased by 2 μm, up to a maximum value of 18 μm (2 μm above the coverslip surface). Thus, the vertical position Z of the probe tip was defined as Z = h_0_ – D. The actuator calibration and the backlash correction were used to estimate the actual position during the whole indentation protocol (see Section 3.2). The indentation depth δ corresponding to each distance D was estimated by taking into account the average thickness of each cell <T_max_> as measured via digital holography (see Section 3.3) and the rest position h_0_:


(1)
$$\delta =<T_{\max}>-Z$$


-The activation of mechanosensitive channels resulted in a transient peak of R as the indicator of intracellular increase in calcium concentration, requiring approximately 40 s to recover the baseline level. Therefore, during the indentation protocol, the R signal was monitored by fluorescence microscopy and a time of 40 s was allowed to pass between two successive indentations in order to appreciate any late activation events.-The value δ at which the signal R overcame a given threshold from the baseline was defined as the indentation at which the cell responded (see Section 3.4 for threshold evaluation): a transient peak in the R signal was related to the influx of Ca^2+^ ions into the cytoplasm, corresponding to the activation of the mechanosensitive channels. On the other hand, a plateau in R signal indicated the rapid decrease of F_340_ and F_380_, i.e., the rupture of the membrane. Every cell was indented through 8 to 18 μm of probe displacement D: if the R peak was observed ahead of cell disruption, the cell was classified as responsive at the corresponding δ. Otherwise, cells were classified as unresponsive.-We defined the activation rate AR as the ratio between the number of responsive cells to the mechanical stimulus and the total tested ones:

(2)$$\mathrm{AR}=\frac{\#\;\mathrm{responsive}\;\mathrm{cells}\;}{\mathrm{total}}$$
Hence, we used this parameter as the measure of the cellular mechanosensitivity for a specific cell group, e.g., A1 WT and A1-Piezo1 (see Section 3.4). For a particular value of indentation depth δ, we defined the activation rate AR_δ_ as the ratio between the number of cells activated at δ and the total tested ones: (3)$${\mathrm{AR}}_{\delta }=\frac{\#\;\mathrm{responsive}\;\mathrm{cells}\;\mathrm{at}\;\delta \;}{\mathrm{total}}$$
We used this parameter to evaluate the mean δ required for mechanosensitive response. 

## 3. Results and Discussion

### 3.1. Stage Calibration

The motion translation stage was calibrated by interferometric measurements (Figure 3a) in order to quantify the actual size of the actuated displacement (Figure 3b: a micrometric motion resolution is required for the assay) and to evaluate the bidirectional positioning accuracy [37]. The expected value for the minimum actuated displacement along the X, Y, and Z directions was estimated to be ~24.4 nm [33]. The actual minimum displacements were measured for the positive and negative direction of the Z-axis and were found as ~21.1 nm and ~18.9 nm, respectively (Appendix A).

#### 3.1.1. Bidirectional Systematic Error

The calibration of the translation stage was performed by actuating 20 displacements of 1 μm forward and backward, repeating this sequence 3 times (Figure 3c). This procedure was conducted to evaluate the actual displacement for each single step and the systematic error of positioning for each directional change in order to correct the deviation due to the backlash. The bidirectional systematic positioning deviation was defined as [37]:(4)$$E=\max\;\left[{\underline{z}}_{i}^{+};{\underline{z}}_{i}^{-}]-\min[{\underline{z}}_{i}^{+};{\underline{z}}_{i}^{-}\right]$$
where ${\underline{z}}_{i}^{+}$ and ${\underline{z}}_{i}^{-}$ are the arithmetic mean of the deviations of the actual position from the target position, obtained by the series of three unidirectional movements to the i-th position along the z-axis, for both the positive and negative approach. For the Z-axis, the bidirectional systematic error with no backlash correction (Figure 3c) was E=3.2 μm.

#### 3.1.2. Backlash Evaluation and Correction

In order to evaluate the backlash of the stage, three series of forward and backward displacements of 100 μm were actuated with the calibration set-up. Naming ${\Delta z}_{i}^{+-}$ the difference between the i-th positioning from the positive and negative approach and ${\Delta z}_{i}^{-+}$ the difference between the i-th positioning from the negative and positive approach, $B^{+-}$ and $B^{-+}$ were considered good estimators for backlash, defined as:(5)$$B^{+-}=\left[{\Delta z}_{i}^{+-}\right]-\left[{\Delta z}_{i}^{+-}\right]≃\;3.7\;\mu m\;$$
(6)$$B^{-+}=\left[{\Delta z}_{i}^{-+}\right]-\left[{\Delta z}_{i}^{-+}\right]\;≃3.1\;\mu m$$
Rounded values of $B^{+-}$ and $B^{-+}$ were used to correct the positioning along the Z-axis (Figure 3d) by overshooting an integer number of steps after changes of direction.

#### 3.1.3. Bidirectional Positioning Accuracy of Axis

In order to characterize the positioning accuracy of the motion stage, the bidirectional positioning accuracy of the axis (A) was evaluated according to ISO guidelines [37]. The quantity A resulted from combining the bidirectional systematic positioning errors and the estimator for axis repeatability of bidirectional positioning with a coverage factor k = 2. Naming $\sigma_{i}^{+}$ and $\sigma_{i}^{-}$ the standard uncertainty of the deviations of actual position from the target as obtained by series of three unidirectional approaches to i-th position along z-axis, for both the positive and the negative approach, the bidirectional positioning accuracy for Z-axis was defined as:(7)$$A=\max\left[{\underline{z}}_{i}^{+}+2\sigma_{i}^{+};{\underline{z}}_{i}^{-}+2\sigma_{i}^{-}]-\min[{\underline{z}}_{i}^{+}-2\sigma_{i}^{+};{\underline{z}}_{i}^{-}-2\sigma_{i}^{-}\right]$$
From calibration of the Z-axis with overshooting for backlash correction (Figure 3d), the bidirectional positioning accuracy obtained was A=2.5 μm.

#### 3.1.4. Uncertainty of Probe Positioning

The uncertainty on positioning Δz was obtained via the sum of the random error E_R_ and the sub-micrometric residual bidirectional systematic error E upon the overshooting correction (E = 0.5 μm). E_R_ was obtained as the cumulative mean deviation z_i_ of displacement sequences in Figure 3c upon backlash correction, resulting in E_R_ = 0.4 μm. Thus, the total uncertainty was Δz = 0.9 μm.

### 3.2. Cellular Morphometry

Cellular response was elicited by the penetration of the probe tip in the cellular membrane. The estimation of the indentation depth δ required a morphological characterisation of the cellular groups, i.e., to quantify the average maximal thickness <T_max_>. Resulting values were obtained by evaluating δ considering the probe displacements D and the initial height h_0_ of the probe tip (h_0_ = 20 μm). The morphometrical characterisation of the A1 cells was provided by digital holographic microscopy (DHM). DHM enabled the recording of 3D information by acquiring interfering wave fronts from a low-power laser coherent light source (635 nm wavelength, 0.2 mW/cm^2^): the laser beam was split into a reference beam and an object beam that passed through the specimen (A1 cells in Locke buffer). A phase shift (Φ) in the object laser beam was yielded by the passage through the cell’s internal structure. By merging object and reference beam, an interference pattern is created and recorded on a digital image sensor (CCD) to numerically reconstruct a phase image, which is displayed and analysed (Figure 4a) [38,39]. The phase shift (Φ_i_) information contained in the i-th pixel is related to optical thickness L_i_ according to:(8)$$\Phi_{i}=\frac{2\pi }{\lambda }\cdot L_{i}=\frac{2\pi }{\lambda }\cdot n\cdot T_{i}$$
where T_i_ is sample thickness, *n* the average refractive index of the object and λ the incident laser wavelength. Using a priori calibration of the background (acquiring the optical thickness of the medium surrounding the cells), the maximum cell thickness (T_max_) was obtained from Equation (8), analysing the maximum phase shift within the area of single cell:(9)$$T_{\max}=\frac{\lambda }{2\pi }\cdot \frac{\underset{i}{\max}\Phi_{i}}{n_{c}-n_{m}}$$
where n_c_ and n_m_ are the mean integral refractive indices of cells and surrounding medium. Reference values of n_c_ = 1.38 and n_m_ = 1.34 were considered based on precedent works [30,40,41,42]. Small deviations in the refractive index of the cell with respect to the adopted value contributed to the overall uncertainty of the measurement by a minimal amount. The A1 WT (45 cells) and A1-Piezo1 (65 cells) were separately analysed to assess average maximal thickness <T_max_> (Figure 4b), and we found no significant variation as a function of Piezo1 expression. In fact, morphometric characterisation resulted in a (9.6 ± 0.3) μm average maximal thickness for WT A1 and (9.2 ± 0.2) μm for A1-Piezo1 cells.

### 3.3. Microindentation: Data Acquisition

To avoid any damage to the cells under investigation, we allowed a maximum experimental time of 50 min. The probe positioning procedure was repeated for each cell and it took about 5 min for each indentation, limiting the number of cells that were effectively probed in a single session to about 10 per dish. Data of the intracellular fluorescence intensities from each channel and fluorescence ratio were extracted with ImageJ software (ROI tool). The normalized fluorescence ratio R was related to intracellular Ca^2+^ concentration: since the activation of mechanosensitive channels provokes the influx of Ca^2+^ ions into the intracellular environment, a transient peak in F_340_ and a simultaneous dip in F_380_ (transient peak in R signal plot) were observed. On the other hand, the rupture of the cell membrane yielded the diffusion of the dye in the surrounding medium and was associated with a rapid decrease of both fluorescence signals (plateau in R plot). The analysis of the recorded fluorescence ratio R signal (see Section 3.4) allowed for the identification of signal peaks elicited by channels opening, enabling us to classify tested cells as responsive or unresponsive and to calculate the activation rate, AR. By observing temporal evolution of R and plotting probe displacements Z against time (Figure 5a,b), the indentation depth δ eliciting cellular response was identified.

### 3.4. Data Analysis

In order to discriminate responsive and unresponsive cells, the time tracking of the fluorescence ratio R was compared to a cell line specific threshold R_T_: the protocol to evaluate the activation threshold was further be extended to different cell lines. We tested if the choice of R_T_ would affect the results of the inter-group behaviour with regard to activation rate AR, defined in Equation (2). We evaluated how AR changed while modifying the threshold between R_T_ = 0 and R_T_ = 4.5 (see Figure 6a). The normalised fluorescence ratio R was analysed with an adapted Python script (SciPy library, the find.peaks tool [43]), and required a previous characterisation of a significant R peak representing “activation”, which we defined as a responsive cell presenting a transient R signal peak that was related to a peak in F_340_ signal and a dip in F_380_. A peak of R signal was correctly classified as an “activation signal” when R > R_T_ and when a peak and a dip occurred in F_340_ and F_380_, respectively (Figure 5c). As a further constraint, a minimum width (at baseline) of w = 30 s was considered for the peak/dip. We found that (Figure 6a) when choosing the value of the threshold to be higher than R_T_ = 3.5 (AR minima in each group), no cell responded to the mechanical stimulus, indicating that this value was too high, while for values lower than R = 1.5 (AR maxima in each group), the signal started to be strongly dependent on the measurement noise and several cells would be marked as responsive (Figure 6a). Furthermore, the validity of this selection was confirmed by the AR plot of the control groups in Ca^2+^-free buffer (Figure 6a). Since mechanosensitive channels opening led to an influx of external Ca^2+^ ions, no increase of R signal was expected by the mechanosensitive activation of cells in Ca^2+^-free solution. Hence, the lower limit of the range was well defined by observing the rapid decrease of AR for Ca^2+^-free groups, which was AR = 0% for R_T_ > 1.5, as expected; meanwhile, a non-zero AR was observed for R_T_ < 1.5. Moreover, between R_T_ = 1.5 and R_T_ = 3.5, the AR demonstrated consistent inter-group behaviour regardless of the chosen specific threshold. Thus, we selected the median of the interval R_T_ = 2.5 as the proper activation threshold to calculate the AR. Figure 6b presents the AR values for each cellular group. It is interesting to see that AR grew as a function of the expression level of Piezo1 (18% and 32% of activation for A1 WT and A1-Piezo1 respectively), whereas cells in Ca^2+^-free medium were totally unresponsive, further suggesting that the increase of cytosolic calcium concentration was only due to an extracellular calcium influx mediated by membrane channels. Moreover, incubation with the mechanotoxin GsMTx4 resulted in a reduction of AR to 7%, confirming the role of mechanosensitive channels in the observed signal. 

As previously anticipated, we found that activation events could be associated with the indentation imposed on the cell. In fact, the activation-triggering probe displacement D was translatable to cell indentation depth δ, as seen in Equation (1). Based on this consideration, we can also plot the activation rate AR_δ_ (% of responsive cells) as a function of the indentation depth δ. Figure 6c shows this representation for A1-Piezo1 and A1 WT cells.

## 4. Conclusions

In this paper we present a method for the evaluation of the mechanosensitivity of living cells associated with the activation of calcium-permeable ion channels of the Piezo family. Our approach is based on the integration of a 3D-printed manipulator with a calcium imaging set-up. The proposed device does not require expensive or complicated components and can be easily implemented in a standard biological laboratory, without the need for specific technical skills. We propose an operational protocol to indent single cells and measure the activation rate as an indicator of whole cell mechanosensitivity. The effectiveness of the indicator was evaluated on cells with different levels of expression of Piezo1 and on wild-type cells, treated with either a mechano-toxin or in calcium-free buffer solution. The results show that the activation rate offers a robust representation of the mechanosensitivity of the cellular population, defined as the probability that any cells will respond to a fixed mechanical stimulus. Moreover, we suggest that the proposed approach can be extended by measuring the indentation depth to which cells respond, which provides finer information on the sensitivity to mechanical stimuli. The activation rate and the sensitivity to indentation convey slightly different information on the cellular population under examination. We expect the activation rate to correlate with the total number of available mechanosensitive ion channels over the cell surface, while the sensitivity to be impacted by clustering phenomena that would simultaneously reduce the activation rate and increase the sensitivity. The importance of the distribution of piezo channels in the plasma membrane has recently been emphasised [44,45], and the tool proposed in this paper could further support the scientific community in the investigation of this elusive mechanism.

## Figures and Tables

**Figure 1 nanomaterials-12-02691-f001:**
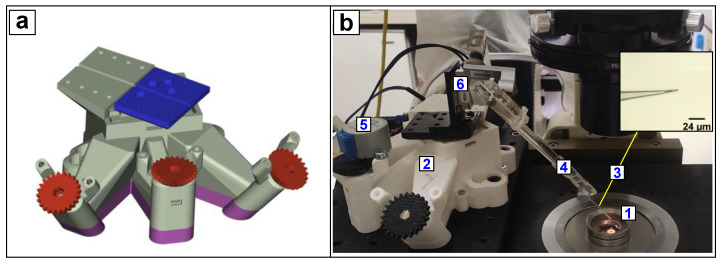
Translation motion stage integrated with microscope set-up for fluorescence calcium imaging. (**a**) Colour-coded CAD of motion translation stage, 3D printed in few parts: a base (purple), a main body (grey), the moving platform (blue) and the actuating screws with knobs (red). (**b**) Cells plated on a glass coverslip held by a steel chamber (1) positioned upon objective of the inverted microscope. The stage (2) enables the driving of Z-axis displacements of the glass probe (3) toward the glass coverslip for single-cell indentation. A micropipette holder (4) is used to hold a glass probe and it is connected to the moving platform of the stage. Vertical movements for indentation are automated by 28-BYJ48 step motor (5). A Newport Agilis AG-LS25 linear piezo motor (6) has no role in the indentation session; it only drives large vertical movements of the probe for initially positioning the steel chamber upon the objective, avoiding chamber edges that would damage the probe tip.

**Figure 2 nanomaterials-12-02691-f002:**
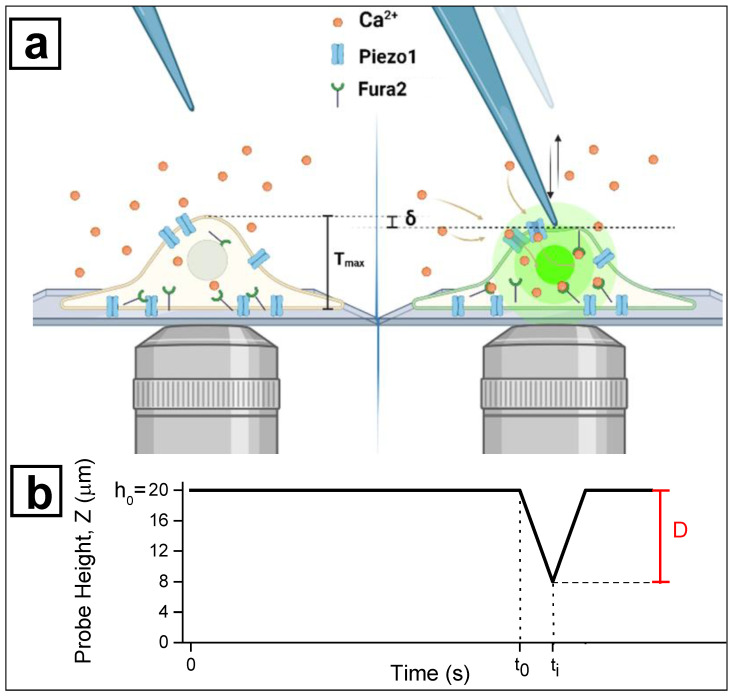
Schematic picture of the single-cell microindentation experiment. (**a**) The perturbation of the cell membrane provokes the opening of the Piezo1 channels, inducing an increase of the intracellular Ca^2+^ concentration, which is monitored by the ratiometric fura2-based fluorescence microscopy. (**b**) Timeline of glass probe position Z: the needle tip is initially positioned 20 μm (h_0_) above the adhesion plane; cell indentation takes place upon an area of maximum thickness T_max_ corresponding to the nucleus zone. At instant t_0_, the probe begins vertically move downward by a distance D; at the instant t_i_, the cellular membrane is indented until a depth δ. Since the moving speed of the motion stage is v = 3 μm/s, the time interval Δt = t_i_ − t_0_ depends on the probe displacement D, which is Δt = 4 s and D = 12 μm in the figure.

**Figure 3 nanomaterials-12-02691-f003:**
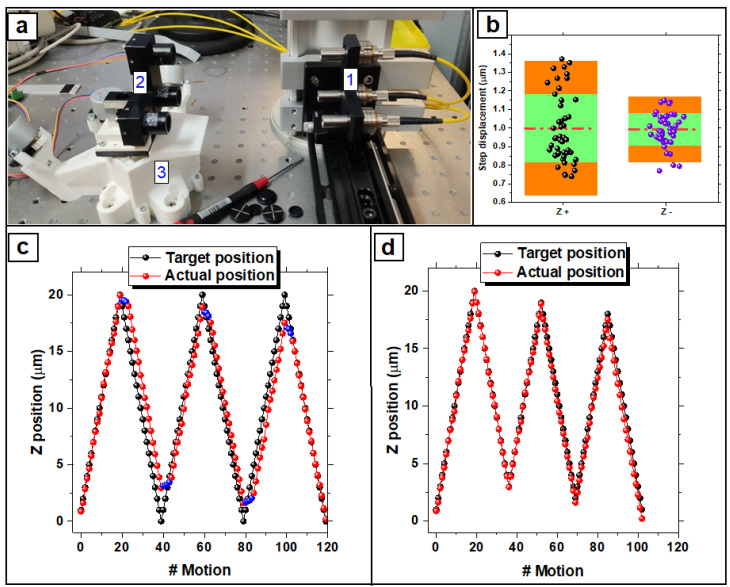
Calibration of the translation stage: step size is evaluated by interferometric measurement. Actuate 20 steps forward and 20 steps backward with nominal displacement of 1 μm per step along Z direction, repeating this displacement sequence 3 times to quantify the backlash and bidirectional positioning accuracy of Z-axis. (**a**) Calibration set-up: 3 fixed IR lasers (1) and three retroreflectors (2) jointed to the moving platform of the stage (3). The interferometer registers actual displacements of the translation stage for three main directions (X, Y and Z) depending on the geometry (in the picture: Y-axis geometry). (**b**) The actual displacements of every step in Z-direction are evaluated for positive (Z+) and negative (Z−) displacements. The red line states the mean of single displacement (1 μm), the green area is limiting steps within one standard deviation and the orange area represents the confidence interval of 95%. (**c**) Three sequences of displacements along the Z-axis: actual positions of retroreflectors jointed to the translation stage (red points) are strongly affected by backlash (blue points): the bidirectional systematic error is E = 3.2 μm. Backlash correction is actuated by overshooting steps to reduce deviation from target positions (black points). (**d**) Conduct 3 sequences of displacements along the Z-axis with backlash correction: the systematic bidirectional error is E = 0.5 μm.

**Figure 4 nanomaterials-12-02691-f004:**
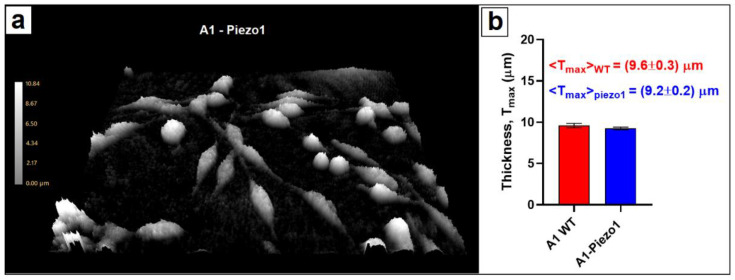
Cellular morphometry via digital holographic microscopy. (**a**) A 3D phase image reconstruction of A1-Piezo1 cells. The scale (at left) relates the phase shift to the thickness (in μm). (**b**) Assessment of the average maximal vertical thickness for A1 WT (*n* = 45) and A1-Piezo1 (*n* = 65) shows no significant morphological differences.

**Figure 5 nanomaterials-12-02691-f005:**
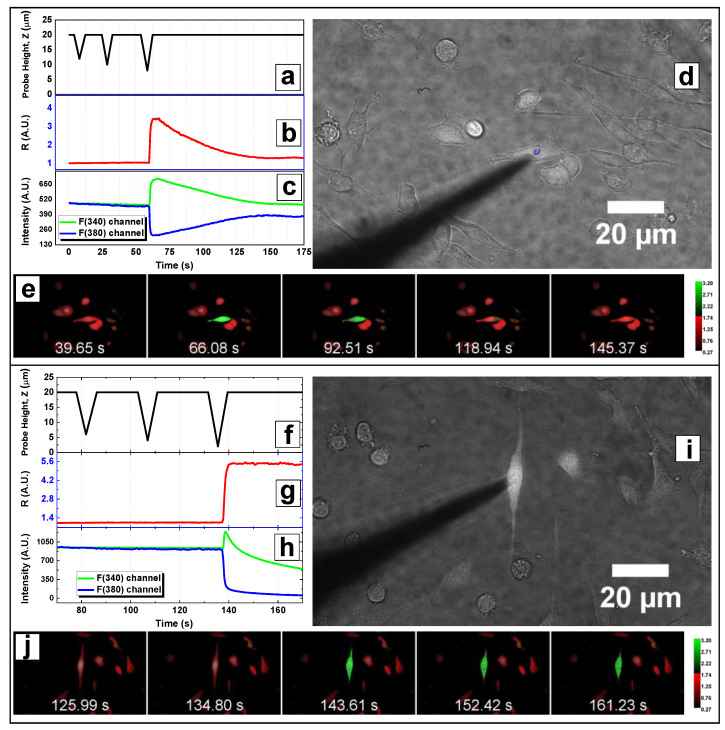
Single-cell microindentation of Piezo1-A1 cells: the changes of intracellular Ca^2+^ concentration are monitored using ratiometric fluorescence microscopy; examples of a responsive (**a**–**e**) and an unresponsive cell (**f**–**j**) are reported. Cell-indenter glass probe is initially positioned above (h_0_ = 20 μm) cell adhesion plane. Thus, deeper and deeper vertical displacements are actuated (**a**,**f**) until contact with cell membrane. Fluorescence intensity in respect to the excitation wavelengths of 340 nm and 380 nm is averaged over the intracellular area, and displayed in a plot of the fluorescence ratio R (**b**,**g**) or a channels plot (**c**,**h**). Both responsive and unresponsive cells are indented until membrane rupture occurs, resulting in a plateau in the R plot. Before the disruptive indentation, the cell presents a transient peak in R signal (**b**) with tFWHM~40 s (responsive). On the other hand, when no increase in the trace of R is observed before the plateau, i.e., the membrane rupture, cell is classified as unresponsive (**g**). The initial positioning of the probe requires a tip position visualisation: bright-field microscopy is used to monitor the shadow of the tip (**d**,**i**), which can be moved above cell area through horizontal displacements of the probe. Fluorescence ratio R is imaged (Appendix A) on five representative color-coded frames displaying the peak of a responsive cell € and the plateau of a ruptured unresponsive cell (**j**).

**Figure 6 nanomaterials-12-02691-f006:**
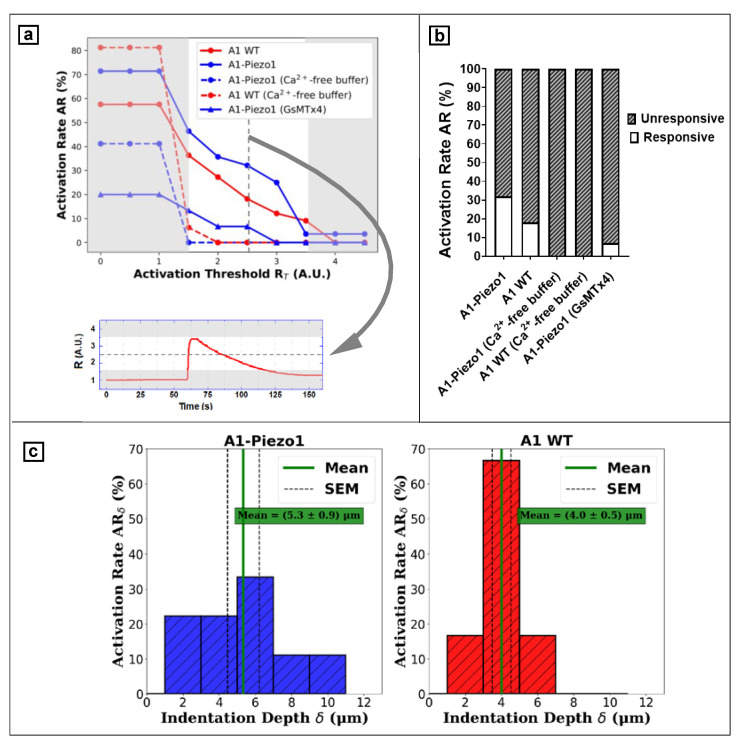
Data analysis and device validation. Assessment of cell responsivity requires setting a proper fluorescence ratio threshold R_T_. Among activated cells, average indentation depth δ obtained from each group provides a calibration parameter. (**a**) Relationship between activation rate AR and the activation threshold R_T_ used for classification; extreme values of threshold R_T_ (grey area) were excluded to avoid misleading values of AR, and the median R_T_ = 2.5 was selected for evaluating inter-group AR. (**b**) AR among different A1 cell groups. (**c**) Histogram showing the depth of indentation δ associated to the response event rate AR_δ_ for A1-Piezo1 and A1 WT.

## Data Availability

All details of the analysis are presented in the paper. The three-dimensional drawings for the positioner were based on opensource projects and the original CAD files can be retrieved from the cited references. Any additional information will be provided by the authors on request.

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
