# Peer review of "Cellular Mechanosensitivity: Validation of an Adaptable 3D-Printed Device for Microindentation"

_nanomaterials, 2022, doi:10.3390/nano12152691_

Round 1

Reviewer 1 Report

The authors presented an approach to stimulate single adherent cells through controlled microindentation using a 3D printed actuation platform. The authors systematically calibrated and tested their platform by measuring the meachanosensitivty of a neural mice cell line. This study provides an important tool for the study of mechanotransduction of single cells. However, the presentations of the work need significant improvement before it can be accepted for publication.

Specific comments:

1. Instead of using the name of the cell line used in this study as the title of subsection 2.1, I suggest to replace it with “Cell Culture”. More details regarding the transfection and antibiotic selection experiments should be provided if they have not been elaborated and published somewhere else.

2. In Section 2.2, authors wrote that A1 cells were plated at a density of 50%. What does “a density of 50%” mean?  

3. Section 3.1 “Microindentation Assay” describes how the microindentation assay was conducted. It looks more appropriate to move this part to Section 2 “Materials and Methods”.   

4. To be consistent, all the equations should be labelled with equation numbers so that it is easier to be referred to in the main text.

5. The equation in Section 3.2.1 seems to have typos as one of the arithmetic mean of the deviations is missing.

6. Figure 3b is never cited in the main text.

7. The notations used in the first question in Section 3.2.2 are not consistent with the way how they are defined.

8. Authors obtained value of the bidirectional positioning accuracy A = 2.5 microns, but it not clear how this piece of information is used and what argument the authors would like to make (acceptable or better than others?).

 9. Authors characterized the average maximal thickness for WT A1 and A1-Piezo1 cells using digital holographic microscopy. Is this technique well calibrated? Can authors validate their results by measuring the cell thickness using another technique, for example confocal microscopy?

10. In Figure 5, authors show that a responsive cell exhibit a transient peak of R signal due to the opening of mechanosensitive channels while an unresponsive cell exhibit a plateau in the R plot due to the rupture of the membrane. Can authors further elaborate why the opening of mechanosensitive and the rupture of membrane lead to a sharp increase in the R signal? The fundamental mechanism should be explained clearly in the manuscript.

11. The way how the range of AR values (1.5-3.5) was determined sounds very subjective. Can authors calibrate their rationale by calibrating their results with a more definitive way to determine whether a cell is responsive or not?

Reviewer 2 Report

In this manuscript, Capponi et al. set up a mechanosensitive experiment using a customised 3D-printed actuator. The system was tested using neural mice cell lines to test and scale their system. According to the authors, they measured a difference in cell indentation response.
While I like customised and cheap experimental setups, which opens to more research labs possibilities to perform experiments. Unfortunately, this study failed to convince me that their claims are correct or reasonable.
First, there seems no evidence or proof that the indentation needle indented the cell before an activation was measured, and thus making the estimated indentation depth unreliable. Or in other word, the authors cannot show a difference between a height D that is either not touching the cell or indenting a cell without response. Only when a response is measured the D give evidence that the cell was indented.   
Secondly, the final derivative and comparison of the indentation depth underlies so many systematic errors of various estimations and setup-limitations that there is no significance between the two cells, WT and Piezo1.
Besides this mayor problem, the manuscript is partly unclearly written, and the handling of parameters is quite poor. Some of the parameters are not well introduced and defined. Other only a reference to another section was shown, which is often insufficient to understand the following paragraphs.
Unless I misunderstood some parts of the manuscripts completely, I cannot recommend  the publication of this work.

[Sec. 3.1] The authors explain the general approach and measuring protocol. That is well written when comparing Fig. 2, however, from line 198 it gets unclear (for me).
First, how is the fluorescence ratio R defined? How long was it measured and compared to what... the background intensity in the medium? Along that line in continues with the next two points from lines 201 and 203.
I strongly recommend to add a simple scheme to Fig. 2, where a time line together with the introduced variables are defined. Referring to the future to come sections is not sufficient to understand what is written here. The language is partly not precise enough, which makes it even harder to follow. For example: "For a specific value of indentation,..." I guess the authors mean "For a specific indentation height/depth δ,..." How is it defined? What gets activated?
I understand that this section should give a kind of general overview, but either one keeps it general or explains it specific so that one understands it without reading the other sections first. So I would recommend to keep the precise explanation and adjust the last three points, where R and AR in respect to time are explained more clearly.

[Fig. 4 and Eq. 1.] As the phase shift ? linearly correlated to the sample thickness T, I would recommend to show not only Fig. 4a, but also show a rescaled image of the sample thickness T, so one gets an overview of the estimated cell topology.

[lines 313ff] Again R and AR is used and still not properly introduced. R and D refers to Fig. 5a, but should be Fig. 5a) and b). Further, Fig. 5a should be identified as D, rather than writing "Probe height"... Same for Fig. 5f).
As the first two indentations do not show a response in R, how did the authors actually ensure that they indented the cell already. The authors write "responsive or not-responsiv", but as only a mean D is assumed as the first approach and successively increased, the third indentation might be simple the first cell-contact. How can a "non-responsive cell" with a "non-cell-contact" be distinguished?  

[Figures] All figures should be labeled with the physical parameter that are used and introduced in the manuscript. It makes it hard to read and understand when the reader has to switch between both.

[lines 345ff] I would recommend to not only write always about "values" and show values that have not even a unit. Better to persistently write "R=1.5" rather than 1.5. It is easier to follow. Or alternatively instead of writing "value" rather writing "ratio" in this context.

[lines 354ff] The authors write "It is interesting to see... but I fail to see the message. It is also unclear why Ca-free buffer gives any relevant results, what is the physiological relevance?
When the medium is Ca-free, how does that relate to the 'background' intensity of the medium?

[Fig. 5fgh] Why did the authors not show a later times beyond the 170s, as it is shown for  Fig. 5abc... more than 100sec after 'activation' is shown. And more importantly, why does the green signal decreases below the initial 'background' intensity of the medium?  

[Equation line 362 and Figs. 6c] As the authors cannot (1.) ensure that the height of each cell is <Tmax> resulting in a first error and (2.) that previous approaches indented the cell membrane already, this equation cannot be assumed nor verified, as the at least two parameter (T and D) are not well-defined. The results that are shown in Fig. 6c, as far as I can see, are also not statistically significant, since the estimated indentation death can only be 'obtained' with an error of +/-2 micron, D has an accuracy of only 2 micron, <Tmax> is estimated and average value that does not account of each individual cell, and the experimental precision of placing the micro indentation needle at the position Tmax is far beyond the pixel-resolution that is used to calculate <Tmax>. That are only a few reasons why I do not believe in a significant difference between the A1 Wild Type and Piezo1 cells. There might be one, but with the current approach it is not reasonable to claim.

Watching the movies in the supplemental material one can see that the needle might have touch the cells (nonresponsive_*.avi), why do the authors do not choose the ROI at the poking position. There, the intensity values should be sufficient to extract the first successful poking and relative indentation with a response. See rough example-analysis of the movie using two different approaches, a ROI (time vs. intensity) and a replicing of the cell thickness at the poking position.
Does the cell dye afterwards, when the red-color does not 'recover'? I may have missed that part?!

In the other movie that is shown (responsive_*.avi), it seem that the needle continues to poke after the first response... and after a few indentation the cell seems to die. In Fig. 5 there is no continuation of the indentation approach seen, so why is it different?

Minor comments:
[line 112] "die" => "dye"
[Fig. 4b] the units (micron?!) in the labels are not readable.  
[line 314] (see 3.5) => (see Section 3.5)
[line 323] for readability I recommend either the use of 'non-responsive cell' or 'unresponsive cell' throughout the manuscript
[line 342] What is a 'dedicated Python script'? (dedicated???)

Round 2

Reviewer 1 Report

The authors have fully addressed my comments, so I recommend acceptance for publication as it is. 

Reviewer 2 Report

All of my comments and suggestions have been adressed.